# Human Spinal Motor Neurons Are Particularly Vulnerable to Cerebrospinal Fluid of Amyotrophic Lateral Sclerosis Patients

**DOI:** 10.3390/ijms21103564

**Published:** 2020-05-18

**Authors:** Stefan Bräuer, René Günther, Jared Sterneckert, Hannes Glaß, Andreas Hermann

**Affiliations:** 1Department of Neurology, Technische Universität Dresden, 01307 Dresden, Germany; stefan.braeuer@uniklinikum-dresden.de (S.B.); Rene.Guenther@uniklinikum-dresden.de (R.G.); 2Department of Neurology, Städtisches Klinikum Dresden, 01129 Dresden, Germany; 3German Center for Neurodegenerative Diseases (DZNE), 01307 Dresden, Germany; 4Center for Regenerative Therapies Dresden (CRTD), Technische Universität Dresden, 01307 Dresden, Germany; Jared.Sterneckert@tu-dresden.de; 5Translational Neurodegeneration Section “Albrecht-Kossel”, Department of Neurology, University Medical Center Rostock, University of Rostock, 18147 Rostock, Germany; hannes.glass@med.uni-rostock.de; 6German Center for Neurodegenerative Diseases (DZNE) Rostock, 18147 Rostock, Germany

**Keywords:** amyotrophic lateral sclerosis, ALS, cerebrospinal fluid, Golgi fragmentation, superoxide dismutase 1, fused in sarcoma

## Abstract

Amyotrophic lateral sclerosis (ALS) is the most common and devastating motor neuron (MN) disease. Its pathophysiological cascade is still enigmatic. More than 90% of ALS patients suffer from sporadic ALS, which makes it specifically demanding to generate appropriate model systems. One interesting aspect considering the seeding, spreading and further disease development of ALS is the cerebrospinal fluid (CSF). We therefore asked whether CSF from sporadic ALS patients is capable of causing disease typical changes in human patient-derived spinal MN cultures and thus could represent a novel model system for sporadic ALS. By using induced pluripotent stem cell (iPSC)-derived MNs from healthy controls and monogenetic forms of ALS we could demonstrate a harmful effect of ALS-CSF on healthy donor-derived human MNs. Golgi fragmentation—a typical finding in lower organism models and human postmortem tissue—was induced solely by addition of ALS-CSF, but not control-CSF. No other neurodegenerative hallmarks—including pathological protein aggregation—were found, underpinning Golgi fragmentation as early event in the neurodegenerative cascade. Of note, these changes occurred predominantly in MNs, the cell type primarily affected in ALS. We thus present a novel way to model early features of sporadic ALS.

## 1. Introduction

Amyotrophic lateral sclerosis (ALS) is the most common and devastating motor neuron (MN) disease. Its initial cause and the mechanisms of further progression are still enigmatic nowadays. The cerebrospinal fluid (CSF) seems to play a major role within the deathly cascade, which is particularly true concerning pathological protein aggregates and their spreading across the central nervous system [1,2].

Several studies have demonstrated diverse pathological effects of ALS-CSF in different in vivo and in vitro models [3,4,5,6,7]. These models mainly include primary murine cortical or spinal MN cultures and spinal cord neuroblastoma hybrid cell line NSC34 cultures. A multitude of phenotypes could be observed by this including aggregation of transactive response DNA binding protein 43 kDA (TDP-43) in the case of ALS-frontotemporal dementia (FTLD)-CSF [8], neurofilament abnormalities [9], gliosis [10], endoplasmic reticulum (ER)-stress [6], mitochondrial dysfunction [11] and Golgi fragmentation [6,12]. In addition, intrathecal infusion of ALS-CSF in an in vivo rat model mimicked neuropathological changes similar to the ones found in postmortem tissue of ALS patients [4]. Furthermore, interesting is the fact that not only CSF but also extracellular mutant superoxide dismutase 1 (SOD1) itself was able to induce Golgi fragmentation and the inhibition of ER-Golgi trafficking [13]. The term Golgi fragmentation describes a loss of the intactness of this cisternal structure and its dispersion across the cell [14,15]. However, nothing is known about how ALS-CSF affects human patient-derived induced pluripotent stem cell (iPSC)-derived MNs, currently believed to be the gold standard for in vitro disease modeling of ALS. Additionally, effects on typical ALS aggregating proteins are barely reported.

Golgi fragmentation in MNs themselves has been reported in different in vitro and in vivo ALS models [16,17]; reviewed by Sundaramoorthy and colleagues [15]. It is also present in postmortem sections of ALS patients [18,19,20,21,22,23,24]. Moreover, Golgi fragmentation is reported in several other neurodegenerative diseases like Alzheimer’s disease and Parkinson’s disease [25,26,27,28]. There is convincing evidence that Golgi fragmentation is an early event in the disease process [23,26,29,30] and not only a consequence of apoptosis [30] but rather inducing it [26]. Thus, similar to other neurodegenerative disease conditions [31], Golgi fragmentation in ALS could be an event that takes place before seeding of the aggregates, axon degeneration and manifestation of clinical symptoms occurs [30,32,33].

The main goal of our study was to establish a new model system for sporadic ALS by investigating the ability of CSF derived from sporadic ALS patients to induce motor neuronal degeneration in the current gold standard cell culture model of ALS, namely patient specific iPSC-derived MNs.

## 2. Results

### 2.1. Patient Characteristics

We used human iPSC-derived neural progenitor cells (NPC) from healthy subjects [34,35,36,37,38]. These cell lines were genotyped for most common monogenetic causes of ALS (namely C9ORF72, SOD1, FUS, TARDBP) excluding presymptomatic donors (female, 50 years; female, 45 years; male, 60 years; female, 48 years). Furthermore, we analyzed the effects on monogenetic ALS cell lines (SOD1-p.R115G, male, 59 years; FUS-p.R521C, female, 58 years; TDP-43p.S393L, female, 87 years). CSF was collected from 19 patients. The ALS group consisted of 11 patients (3 women and 8 men) with a mean age of 63 (SD = 12.2) years. The group of 8 control patients (3 women and 5 men) had a mean age of 47 (SD = 21.4) years (See materials and methods Table 1 and Table 2 for further information).

### 2.2. CSF Affects Proliferation of NPCs

One important observation was that CSF, irrespective of whether from ALS patients or healthy controls, caused a significant proliferation of remaining NPCs in our neuronal cell culture in a clear dose dependent manner, correlating with the amount of CSF and length of exposition. As shown in Figure 1, increasing amount of CSF (10%, 20% and 50%) dramatically increased NPC (GFAP^−^, Vimentin^+^) numbers, while not affecting neuronal morphology. Prolonged incubation time additionally increased non neuronal cell numbers. This effect was still obvious, albeit to a lesser extent, when cells were allowed to differentiate for 10 days prior to being incubated with CSF. Therefore, all further experiments were performed under the latter condition (Figure 1a). To avoid that the effects of CSF are mainly due to lack dilution of standard medium/growth factors we diluted the medium of untreated MNs with PBS and further varied the concentration of the added factors (GDNF, BDNF, cAMP) in the CSF blended medium. However, we found no significant influence of this. Since we used this long-term culture protocol with significant amount of CSF needed, we had to pool the CSF samples from controls and ALS patients in all following experiments focusing on group differences.

### 2.3. No Neuronal Loss or Neuronal Network Degeneration by ALS-CSF

We found a significant influence of CSF treatment itself on the number of neurons relative to total cells (TuJ-1/Hoechst) in control cultures, but no ALS-CSF specific effect. In detail, the amount of neuronal cells treated with either ALS- or control-CSF was significantly decreased compared to non-treated cells (44.95% vs. 69.18%, *p* < 0.05; 45.33% vs. 69.18%, *p* < 0.05; one-way ANOVA with Tukey post-hoc test; Figure 2a). However, no significant difference was found in the comparison of cells treated with ALS-CSF and those exposed to CSF from controls (44.95% vs. 45.33%, *p* = 0.9990 by one-way ANOVA with Tukey post-hoc test). Furthermore, the control cell line showed neither significant difference in the number of MNs relative to total cells (MN/Hoechst, Figure 2b) nor in the number of MNs relative to total neurons (MN/TuJ-1, Figure 2c), when exposed to either control-CSF or ALS-CSF.

When analyzing a monogenetic ALS cell line, namely the SOD-1-mutant cell line (p.R115G), we saw a significant decrease in the number of MNs relative to total cells (MN/Hoechst) when comparing ALS- or control-CSF-treatment to non-treated cells (10.59% vs. 38.24%, *p* < 0.05; 15.59% vs. 38.24%, *p* < 0.05; one-way ANOVA with Tukey post-hoc test) (Figure 2b). Of note, there was no difference between ALS-CSF treatment and control-CSF exposition in this cell line. Furthermore, the treatment with CSF showed a significant influence on the relation of total neurons (TuJ1/Hoechst) (Kruskal-Wallis test *H(2)* = 7.636, *p* < 0.05; Figure 2a). We could detect a significant decrease comparing ALS- and no CSF treatment condition (29.21% vs. 74.33%, *z* = 2.763, *p* < 0.05 with Dunn-Bonferroni post-hoc test), but no difference between ALS- and control-CSF exposition (29.21% vs. 39.91%, *z* = 1.279, *p* = 0.602 with Dunn-Bonferroni post-hoc test).

There was no overt visible difference in the neuronal network comparing cells treated with control-CSF or ALS-CSF (Figure 3). We therefore analyzed the total size of the SMI-32 and TuJ-1 network (Figure 3b) and calculated the network degeneration (Figure 3c). There was no significant difference between either CSF treated condition and, furthermore, no difference when comparing them with the no CSF treatment, neither in the control- or in the mutant-SOD1 (p.R115G) cell line. We could not detect a significant distinction between the control- and mutant-SOD1 (p.R115G) cell line, either.

### 2.4. No Signs of Pathological Aggregate Formation by ALS-CSF

Intrathecal infusion of ALS-CSF in an in vivo rat model caused pathological abnormalities similar to those found in the postmortem sections of ALS patients, namely accumulation of cytoplasmic TDP-43, indicating a spread of the disease via CSF [4]. Further investigations showed the induction of TDP-43 aggregation in the case of ALS-FTD-CSF using a glioblastoma cell model [8]. We therefore asked whether ALS-CSF was able to induce aggregate formation in healthy donor-derived MNs. Following our treatment protocol, there were no signs of significant aggregates containing disease relevant proteins such as TDP-43, fused in sarcoma (FUS), or SOD-1. Additionally, no relevant cytoplasmic mislocalization of TDP-43 and/or FUS was detected in iPSC-derived neurons (Figure 4a–c). We next asked whether ALS-CSF is able to seed aggregation in patient-derived MNs expressing mutations in ALS aggregating proteins, namely TDP43, SOD1 and FUS mutant. Interestingly, by using SOD1 p.R115G (Figure 4d–f), TDP-43 p.S393L (data not shown) and FUS p.R521C cell lines (Figure 4g–i), we could detect neither aggregation of any disease relevant protein nor cytoplasmic mislocalization of FUS and/or TDP-43 (Figure 4).

### 2.5. ALS-CSF Induces Golgi Fragmentation in Patient-Derived Motor Neurons

Golgi fragmentation has been reported in different ALS-models and postmortem sections of ALS patients [18,19,22,23] reviewed by Sundaramoorthy and colleagues [15]. In addition, Golgi fragmentation has been extensively described as an early event in the neurodegenerative cascade [16,17,25,26,27,28]. We therefore investigated whether the addition of patient-CSF is sufficient to cause this early hallmark in healthy donor-derived iPSC-derived motor neurons. After CSF exposition, we classified the Golgi apparatus depending on its structure in four different categories (Figure 5a). As further shown in Figure 5, the application of ALS-CSF to healthy donor-derived MNs caused a significant higher rate of Golgi fragmentation in MNs when compared to cells treated with control-CSF (22.99% vs. 2.35%, *p* < 0.001 by one-way ANOVA with Tukey post-hoc test) or without CSF (22.99% vs. 2.21%, *p* < 0.001 by one-way ANOVA with Tukey post-hoc test). Control-CSF had no effect when compared to the non-treated condition (2.35% vs. 2.21%, *p* = 1.00 by one-way ANOVA with Tukey post-hoc test).

Similar results could be seen in the Golgi fragmentation rates of non-MN-neurons (SMI32−/ TuJ-1+), albeit to a much smaller extent compared to MNs (22.99% vs. 9.15%, *p* < 0.001 by two-way ANOVA with Bonferroni post-hoc test), indicating a higher vulnerability of MNs to ALS-CSF.

In contrast, there was no significant increase of the Golgi fragmentation rates when ALS-CSF was applied to the mutant SOD1 cell line. Of note, the fragmentation rate of not treated mutant SOD-1 neurons was remarkably higher compared to the control cell line.

## 3. Discussion

There is evidence that CSF from ALS patients can directly harm neurons and induce changes associated with ALS pathophysiology. However, data does not exist about such effects on human patient-derived MNs, nor have the effects on modern molecular markers of ALS been well investigated. By using iPSC-derived MNs from healthy controls and monogenetic forms of ALS we demonstrate a harmful effect of CSF from patients suffering from sporadic ALS on healthy donor-derived human MNs. Golgi fragmentation—a typical finding in lower organism ALS models and human post mortem tissue of ALS patients—was induced solely by the addition of ALS- but not control-CSF. Strikingly, these changes occurred predominantly in MNs, the cell type primarily affected in ALS [39,40]. Thus, the presented system might serve as valuable human MN model system for sporadic ALS (Figure 6). Interestingly, this was not obvious in SOD1 p.R115G mutant MNs most likely due to already slightly increased fragmentation rates in the SOD1 mutant MNs. It might also point towards genotype specific effects, which, however, needs to be proven in future systematic studies across a variety of ALS genes and different mutations.

In our study, we did not observe significant neuronal network degeneration, MN-loss or protein aggregates containing SOD1, FUS or TDP-43 (Figure 3 and Figure 4. Indeed, there is a large debate as to whether protein aggregations can be detected in iPSC-derived motor neurons [37], but there is at least clear evidence that it is found in FUS-ALS models [34,35,41]. Thus, even though it might not be surprising that we could not induce any kind of cytoplasmic aggregation of TDP-43 and/or SOD1, it was also not visible in the FUS-ALS model. Another explanation might be the protocol used in the study, which is known to induce structural neurodegeneration only at later stages [35]. Since Golgi fragmentation is expected to be an early event in the pathophysiological cascade [23,26,29,30], it is plausible that it occurs prior to further changes, which could develop either later or under additional stress conditions [34,35,36,37]. Furthermore, ALS-CSF might not be “toxic enough” to induce the complete spectrum of pathological changes seen in postmortem tissue, which is particularly true for inducing pathological aggregates via CSF [8]. Concentration of disease relevant proteins and/or aggregates in patient-derived CSF might be too low to act as agent to induce further aggregation (seeding). This would fit with a recent study by Ding and colleagues using CSF from ALS and additionally ALS-FTLD patients in which only ALS-FTLD-CSF was able to induce TDP-43 aggregates in target cells. It is well known that ALS-FTLD patients have a much more widespread TDP-43 pathology compared to ALS patients without co-occurrence of FTLD [42]. The main differences from our study were the use of a glioblastoma cell line (U251) with obvious differences in general cell metabolism and a longer incubation time with CSF (21 days vs. 6 days). Of note, we additionally did not observe pathological protein mislocalization or aggregate formation when treating monogenetic ALS-patient-derived MNs with ALS-CSF (Figure 4), which are known to have a higher propensity to serve as seeding acceptors [43]. Furthermore, the seeding ability of the mutant protein also differs significantly between different mutations [1,41,44,45,46], which might explain why we did not find seeding events in patient-derived MNs. Since we modelled spinal MNs, we cannot exclude that this might differ in cortical neuronal subtypes. Finally, the employed methods might not have been able to visualize smaller aggregates or oligomers of misfolded proteins. However, we intended to analyze the samples as similar as possible to human postmortem sections in which cytoplasmic protein localization and aggregation is the hallmark pathology.

We could not observe a significant reduction of MNs or neurons in general by exposition to ALS-CSF compared to control-CSF, when healthy donor-derived MNs were treated. Several reasons could explain this difference from results from murine primary cortical/spinal motor neuron cultures [3,47], including human vs. murine cell culture system, medium composition or preanalytic CSF preparation. Another major concern regards the cellular age. Because iPSC-derived neurons do resemble fetal neurons and are not as old as the ones normally affected in ALS patients. Even though all primary neuron cell culture models are derived from fetal brains as well—and thus this cannot be the sole reason for the differences—human development takes much longer, and hence neuronal maturation affects phenotype appearance more severely especially in stem cell derived models [34,35]. The reduction of neurons and MNs by exposition to control CSF with no differences to ALS-CSF (Figure 2a,b) indicated a disease-independent effect on human iPSC-derived neurons. This finding can likely be caused by a reduction of specific medium components in case of CSF-concentrations above 10%. Interestingly, this seems to be similar in adult human neural precursor cells in which CSF promotes stemness [48]. Our data point more towards a decreased relative amount of MNs and neurons compared to total cells due to a proliferative effect on NPCs rather than neuronal cell death.

Our study has some limitations. First, the amount, duration and time point of CSF treatment was limited due to the significant proliferation of NPCs induced by CSF and thus an overgrowth after longer incubation times. This effect was independent of the CSF-type. One additional limitation of our study is the use of pooled CSF. However, this was necessary due to the long time culture conditions and the protocol for human iPSC-derived MNs with the need for medium change every other day (Figure 1). Thus, we cannot answer the question as to whether single ALS patient-derived CSF might vary depending on clinical parameters such as ALS subtype, disease stage or even family background [47]. However, the study rationale was to investigate whether CSF from sporadic ALS patients could induce an MN-like degeneration in the dish, and thus we intended to analyze group differences rather than ALS subtype or individual patient’s specific differences. Furthermore, the model system presented represents one aspect of ALS (namely a model for spinal cord degeneration), and thus there might be differences when analyzing other neuronal cell populations affected, e.g., cortical neurons. The control group were not healthy controls but consisted of patients suffering from non-inflammatory, non-neurodegenerative neurological disorders (e.g., headache), who underwent CSF puncture during clinical routine conditions. However, CSF samples depicting signs of inflammation and/or classical neuropathological markers were excluded (see Table 1). Additionally, we used the marker SMI-32 for the staining of MNs. However, as described in the methods section, following this particular protocol for MN differentiation, all SMI-32 stained MNs were additionally positive for specific motor neuronal markers including Hb9 (Homeobox gene 9), Cholinacetyltransferase (ChAt) and ISLET1 (ISL LH4 homeobox 1) [34,35]. We clearly show, for the first time, that sporadic ALS-patient-derived CSF is sufficient to induce typical signs of neurodegeneration (Golgi fragmentation) and that human iPSC-derived motor neurons, the predominantly affected cell type in ALS, seem to be specifically vulnerable.

## 4. Materials and Methods

### 4.1. Patient Material

We included patient material for (i) CSF derivation and (ii) cell derivation.

Add (i): CSF was taken from 11 ALS patients and 8 control subjects (who suffered from non-inflammatory, non-neurodegenerative neurological disorders) under clinical routine conditions and independent of this study. The use of CSF was approved by the local Ethics committee (EK393122012, EK 49022016). CSF was immediately centrifuged (2100 rpm at room temperature for 10 min) and supernatant was stored at −80 °C until use. We included patients with definite, probable or possible amyotrophic lateral sclerosis, according to the revised El Escorial criteria [49]. Patients suffering from genetically proven spinal muscular atrophy, spinal bulbar muscular atrophy, genetic ALS and FTD overlap syndromes were excluded. Details on patients and CSF characteristics are shown in Table 1.

Add (ii): We included cell lines carrying FUS (p.R521C), TDP-43 (p.S393L), SOD1 (p.R115G) mutations and systematically compared them to four control iPSC lines from healthy volunteers (Table 2). All procedures were in accordance with the Helsinki convention and approved by the local Ethical Committee of the Technische Universität Dresden, Germany (EK45022009, approved 21 April 2009; EK393122012 approved 11 March 2013). Patients as well as controls gave their written informed consent prior to skin biopsy.

### 4.2. Generation and Expansion of iPSCs, in Vitro Differentiation of Embryoid Bodies, AP Staining and Immunofluorescence on iPSC Colonies and Derivation of iPSC-Derived Neuroprecursor Cells

These procedures were performed as previously described by Nauman et al., Japtok et al., and Kreiter et al. [34,35,37].

### 4.3. Motor Neuron Differentiation

The basal medium (N2B27) for the final MN differentiation consisted of 48.75% DMEM/F12 (Invitrogen, Carlsbad, CA, USA), 48.75% Neurobasal medium (Invitrogen, Carlsbad, CA, USA), 1% penicillin/streptomycin/glutamine (Thermo Fisher Scientific, Waltham, MA, USA), 1% B27 supplement without vitamin A (Invitrogen, Carlsbad, CA, USA) and 0.5% N2 supplement (Invitrogen, Carlsbad, CA, USA). NPCs were taken from the liquid-nitrogen-storage (−196 °C), then seeded and expanded using N2B27 medium containing additional 150 µM ascorbic acid (Sigma Aldrich, St. Louis, MO, USA), 3 µM CHIR99021 (Cayman chemical company, Ann Arbor, MI, USA) and 0.5 µM purmorphamin (Cayman chemical company, Ann Arbor, MI, USA). After at least two more passages during NPC expansion, patterning was started after reseeding, using N2B27 medium with 1 µM purmorphamin and 1 µM retinoic acid (Sigma Aldrich, St. Louis, MO, USA). After 9 days cells were re-seeded with a density of 50,000 cells per well (26.316 cells/cm²) on PLO/laminin-coated coverslips and differentiation was started using N2B27 medium additionally containing 10 ng/mL rhBDNF (Promega, Madison, WI, USA), 500 µM dbcAMP (Sigma Aldrich, St. Louis, MO, USA) and 10 ng/ mL rhGDNF (Sigma Aldrich, St. Louis, MO, USA). Following this protocol all SMI-32+ cells correspond to MNs and were positively co-stained for Hb9 (Homeobox gene 9), Cholinacetyltransferase (ChAt) and ISLET1 (ISL LH4 homeobox 1) [35,37]. All further analyses were performed according to this protocol.

### 4.4. CSF Treatment (See also Figure 1a)

Since CSF caused a significant proliferation of NPCs in our cell cultures (Figure 1), we had to find a suitable protocol for our experiments. We therefore analyzed the optimal time point of CSF application. We tested different options (immediately after seeding, 24 h, 48 h and 10 days later) using different concentrations (10%, 20%, 50%) of test-CSF. We had the best results initiating the treatment at day 10 after seeding for differentiation, using CSF concentrations of 10% and 20%. We then evaluated the duration of CSF-treatment (24 h, 48 h, 72 h, 96 h, 6 d) and the optimal concentration (10%, 20%, 50%). For this part of our experiment we started using ALS- and control-CSF. This evaluation led to the decision of treating with 20% CSF for 6 days, the highest possible concentration with no extensive NPC proliferation. For the final experiments, CSF from 11 ALS patients and 8 control patients was mixed in equal parts. CSF treatment was started 10 days after the reseeding, blending the differentiation-medium with CSF in the proportion 1:5. Cells were either treated with 20% ALS-, 20% control-, or no CSF. Since the number of seeded cells was constant throughout all experiments, the ratio of CSF to the initial amount of cells was the same in all experiments. Medium was changed every second day throughout the whole protocol. After 6 days of treatment, cells were fixated for immunofluorescence.

### 4.5. Immunofluorescence of Spinal Motor Neurons

For immunofluorescence staining, cells were washed twice with phosphate-buffered-saline (PBS) without Ca^2+^/Mg^2+^ (LifeTechnologies, Carlsbad, CA, USA) and fixed with 4% paraformaldehyde in PBS for 10 min at room temperature. Paraformaldehyde was aspirated and cells were washed three times with PBS. Fixed cells were first permeabilized for 10 min in 0.2% Triton X solution and subsequently incubated for 1 h at room temperature in blocking solution (1% bovine serum albumin, 5% donkey serum, 0.3M glycine (Carl Roth GmBH & Co. KG, Germany) and 0.02% Triton X-100 (Thermo Fisher Scientific, Waltham, MA, USA) in PBS). Following blocking, primary antibodies were diluted in blocking solution and cells were incubated with primary antibody solution for 12 h at 4 °C. The following primary antibodies were used: chicken anti-SMI-32 (1:10.000, Covance, Princeton, NJ, USA), mouse anti-FUS (1:5000, Sigma Aldrich, St. Louis, MO, USA), rabbit anti-beta-III-Tubulin (1:3000, Covance, Princeton, NJ, USA), mouse anti-GM-130 (1:200, BD Pharmingen, Franklin Lakes, NJ, USA), mouse anti-SOD1 (1:100, Cell Applications Inc., San Diego, CA, USA), rabbit anti-TDP-43 (1:400, Abcam, Cambridge, UK), rabbit anti-GFAP (1:1000 Chemicon, Temecula, CA, USA), mouse anti-vimentin (1:100, Sigma Aldrich, St. Louis, MO, USA). The nuclei were counterstained using Hoechst 33342 (LifeTechnologies, Carlsbad, CA, USA).

### 4.6. Quantification and Statistics

After immunofluorescence staining, cells were counted, and relative amounts were calculated. We took at least 15 pictures of each condition at random fields of view across the respective well. This resulted in from 350 to up to 3000 cells counted in each condition for one individual experiment, which we always performed at least three times. Furthermore, the morphology of the Golgi apparatuses was analyzed using a Zeiss Axio Observer Z1 microscope (Carl Zeiss AG, Oberkochen, Germany) and classified in different categories (Figure 5). One category was defined as Golgi fragmentation (loss of the intactness of the cisternal structure and dispersion of the vesicular compartments across the cell; see Figure 5) and relations were calculated using this number. We analyzed the Golgi apparatus of at least 140 neurons per condition of each individual experiment, which we again always performed at least three times. For the analysis of aggregates containing disease relevant protein such as FUS, TDP-43 and SOD1, we examined at least 200 cells per condition of each experiment, which we performed at least four times. Statistical analyses were done using either one-way ANOVA with Tukey post-hoc test, independent samples t-test, or a two-way ANOVA with Bonferroni post-hoc test using GraphPad prism 7 (Version 7.03, GraphPad Software, San Dieago, CA, USA). In case of non-normal distributed data, analyses were performed using Kruskal-Wallis-test with Dunn-Bonferroni post-hoc test. Analysis of normal distribution was performed using Shapiro-Wilk test. Data are depicted as mean ± SEM from at least three independent experiments each (independent differentiation batches).

### 4.7. Analysis of Neuronal Network Degeneration

We performed the analysis as previously described by Glaß et al. [50]. After imaging a neuron or neuronal network, we converted the picture into a 2-bit image using the “threshold” function from Fiji (ImageJ-win64, open source). Following this, we transformed the obtained mask into a skeletonized network using the skeletonize plugin. A full intact axon without any disruption will yield a ratio of perimeter/section-length of 2. A disrupted axon will have an increased perimeter/section-length of up to 4, when the axon is solely represented by alternating pixel/blank spaces. We then rescaled the value range from 2–4 to 0–1. We normalized the data within each experiment to control-no-CSF condition. For the analysis we took at least 15 pictures at random spots across the respective well of each condition of one individual experiment, which we always performed at least three times.

## Figures and Tables

**Figure 1 ijms-21-03564-f001:**
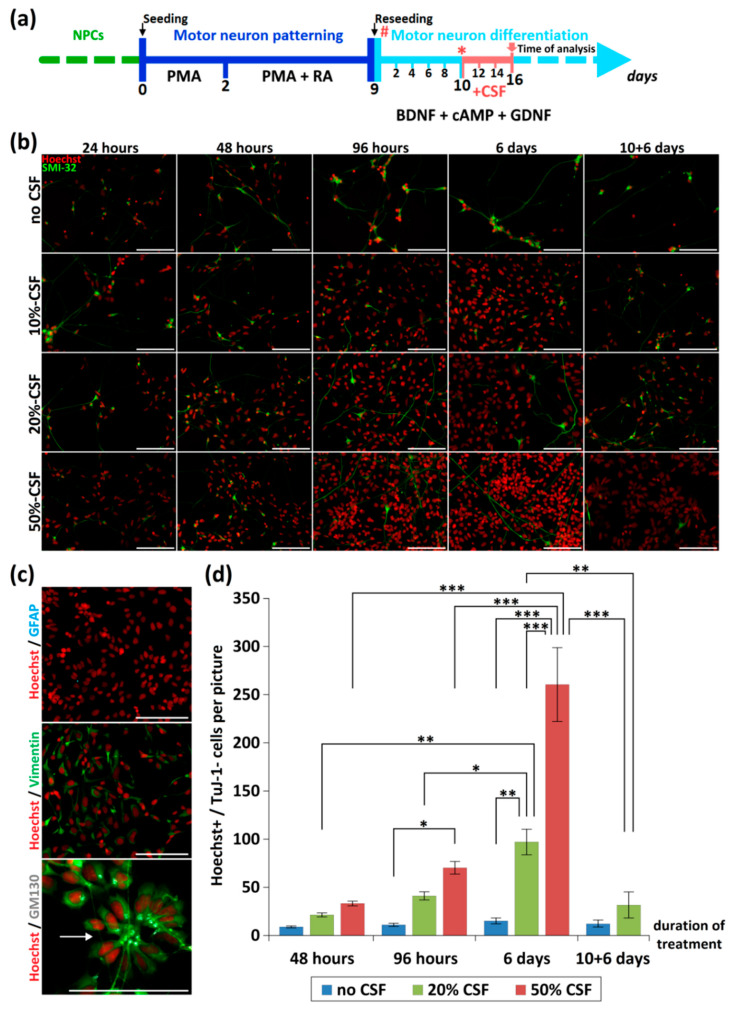
Effect of CSF-application on neurons and NPCs. (**a**) Depicted is the differentiation and treatment scheme. During the initial dose finding experiments, the time point of CSF-application was varied (# and *). For the final experiments CSF was applied 10 days after reseeding (*) and cells were analyzed after 6 days of CSF treatment (red arrow). (**b**) Different concentrations of CSF (0%, 10%, 20%, 50%) were applied to human iPSC-derived MN-cell cultures directly after seeding for 24 h, 48 h, 96 h and 6 days. Additionally, we included a scenario in which we initiated treatment after 10 days of differentiation, followed by 6 days CSF treatment (10 + 6 days). Depending on the duration and concentration of CSF-incubation, a proliferation of NPCs could be seen. (**c**) This effect was independent from the CSF-origin (data not shown) and led to a dramatic overgrowth of GFAP^−^/Vimentin^+^ NPCs, which partially formed neural rosettes (arrow). (**d**) Quantification of non-neuronal cells (Hoechst^+^/Tuj-1^−^) during CSF treatment. */**/*** represents *p* < 0.05/0.01/0.001 as calculated by two-way ANOVA with Bonferroni post-hoc test). Scale bar = 100 µm.

**Figure 2 ijms-21-03564-f002:**
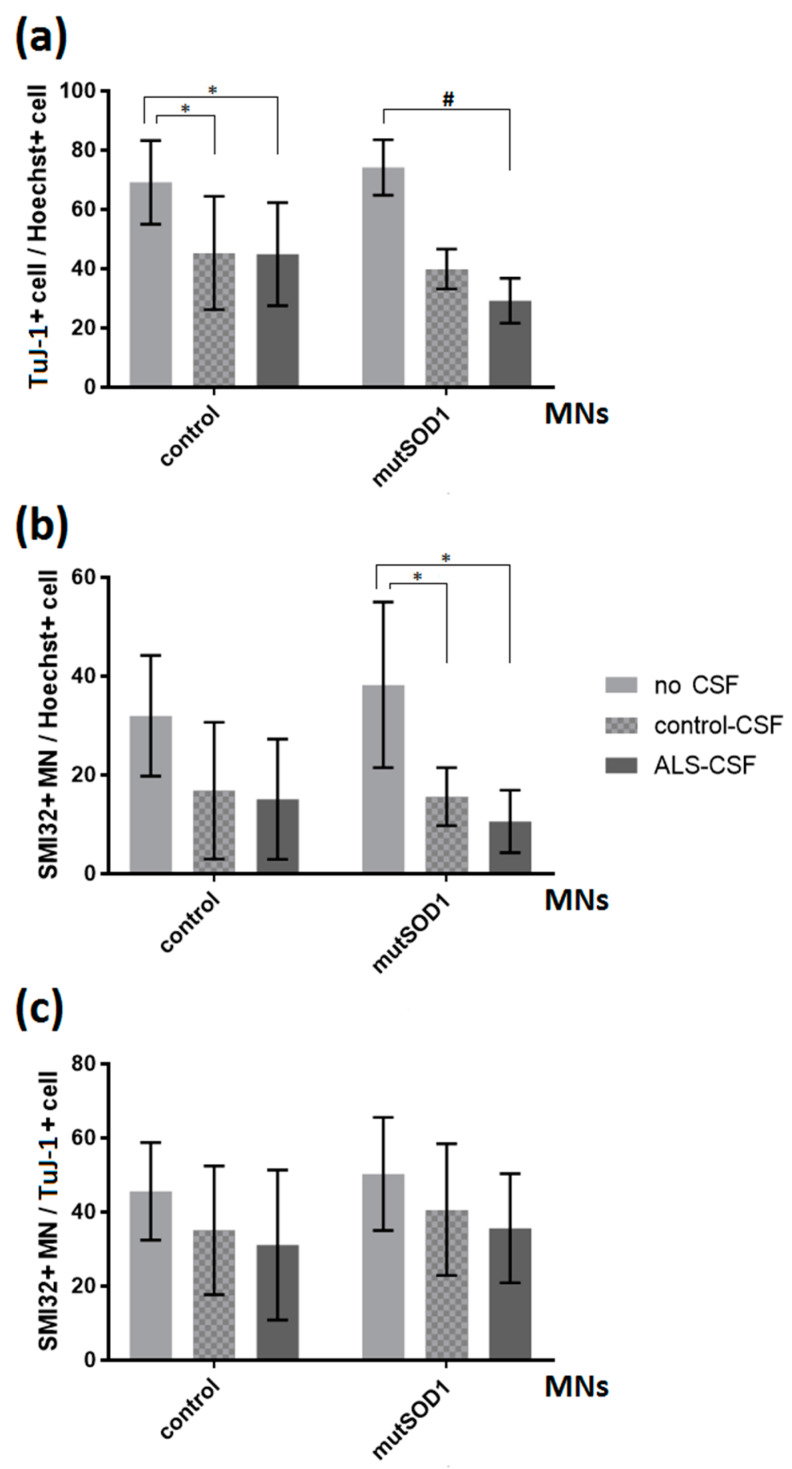
Effects of ALS-CSF on the relations of neuronal cells in control- and mutant SOD1-MNs. (**a**) Left: A significant decrease in the relations of TuJ-1+cell/Hoechst when treated with either control- (* *p*
*<* 0.05 by one-way ANOVA with Tukey post-hoc test) or ALS-CSF (* *p*
*<* 0.05 by one-way ANOVA with Tukey post-hoc test), but no significant effect comparing respective CSF conditions. Right: The mutant SOD1-cell line exhibited a significant decrease in the relations of TuJ-1+cell/Hoechst when exposed to ALS-CSF compared to the no CSF condition (# = Kruskal-Wallis test *H(2)* = 7.636, *p* < 0.05, *z* = 2.763, *p* < 0.05 with Dunn-Bonferroni post-hoc test), but again no significant effect comparing control- with ALS-CSF (29.21% vs. 39.91%, *z* = 1.279, *p* = 0.602 with Dunn-Bonferroni post-hoc test). (**b**) Left: The application of CSF had no significant effect in either cell line analyzing the relation of SMI32+MN/Hoechst in the control-cell line. Right: In SOD1 mutant cells, we detected a significant decrease in the amount of total MNs in relation to all cells (MN/Hoechst) comparing either ALS- or control-CSF-treatment condition and non-treated cells (* *p <* 0.05 by one-way ANOVA with Tukey post-test). (**c**) The application of CSF had no significant effect in either cell line analyzing the relation SMI32+MN/TuJ-1+cell. Data are depicted as mean ± SEM.

**Figure 3 ijms-21-03564-f003:**
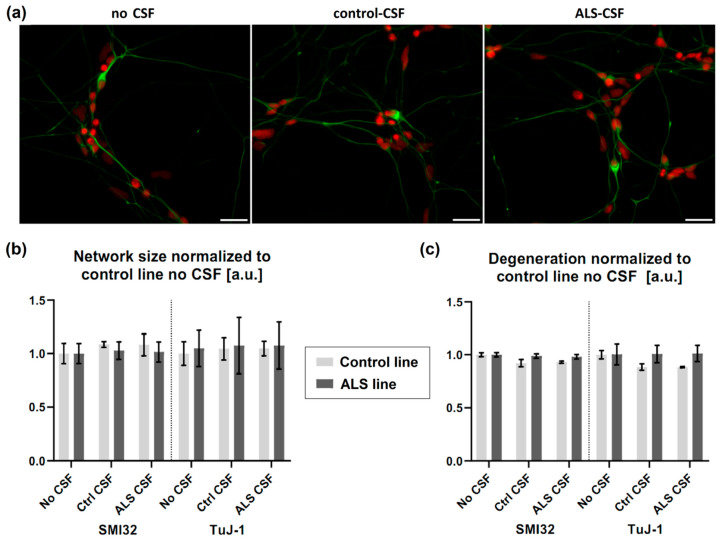
No signs of structural degeneration. (**a**) There was no differing degeneration of the neuronal network (green = SMI32, red = Hoechst) detectable in either cell line treated with CSF. (**b**) There was no significant effect of ALS-CSF on neither the neuronal network size, (**c**) nor network degeneration. Scale bar = 50 µm.

**Figure 4 ijms-21-03564-f004:**
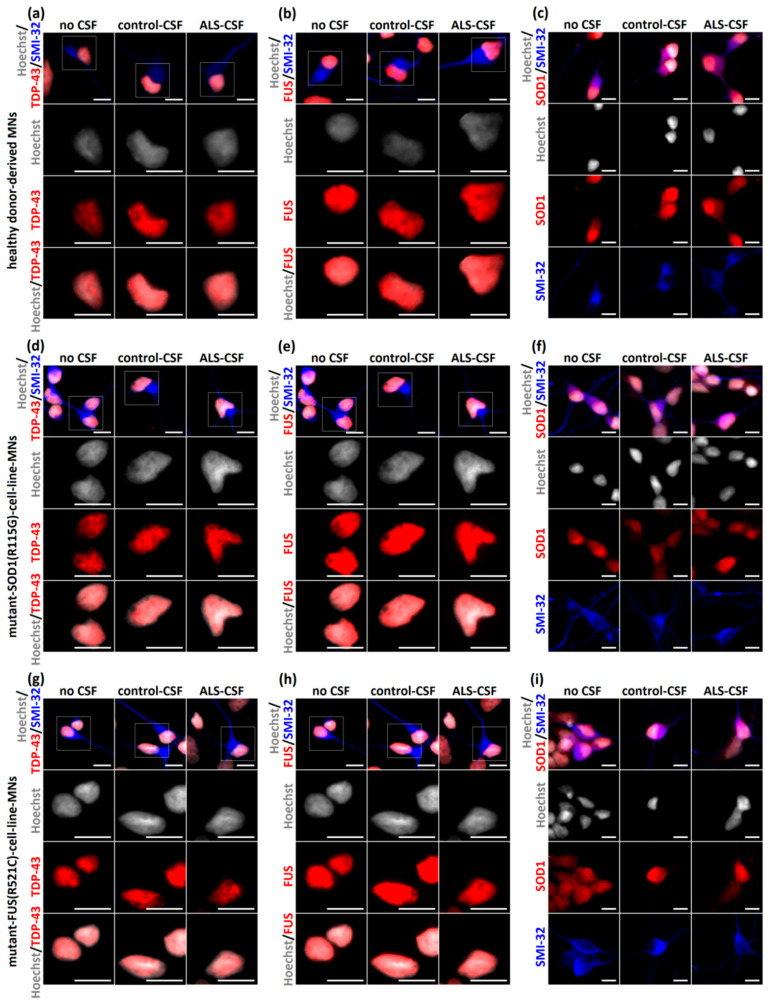
No aggregation of disease-relevant proteins after exposition to ALS-CSF. (**a**–**c**) ALS-CSF caused no detectable aggregates of TDP-43 (**a**), FUS (**b**), or SOD1 (**c**) when applied to healthy donor-derived MNs. Furthermore, there were no differences of cytoplasmic or nuclear localization of any disease relevant protein and—importantly—no cytoplasmic mislocalization of TDP-43 and FUS, respectively. (**d**–**f**) Similar results were detected after applying CSF to the mutant-SOD1-(p.R115G)- and (**g**–**i**) mutant-FUS-(p.R521C)-cell-line MNs. Scale bar = 10 µm.

**Figure 5 ijms-21-03564-f005:**
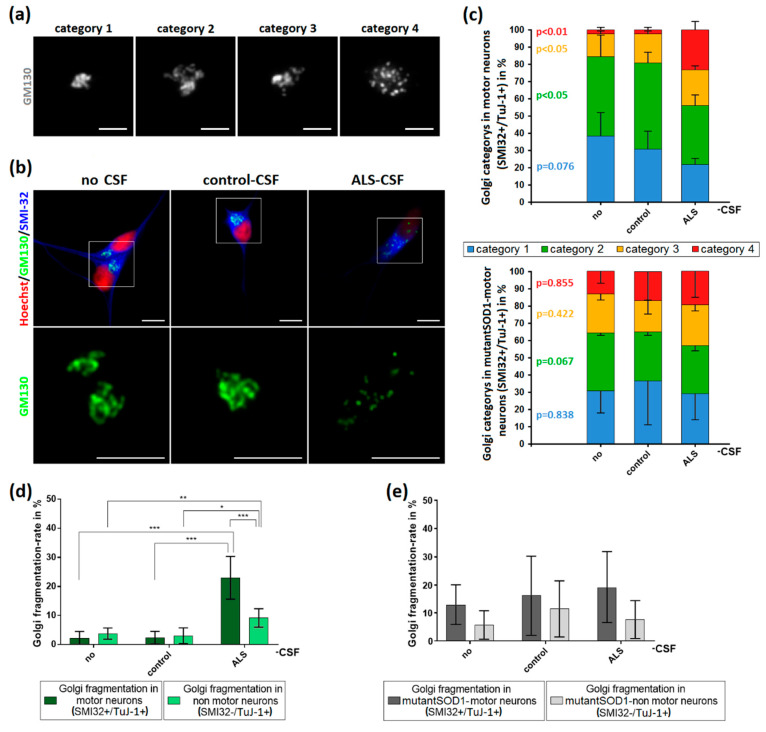
Induction of Golgi fragmentation predominantly in MNs by the application of ALS-CSF. (**a**) For analysis, Golgi complexes were categorized depending on their structure. Category 1 was defined as small, compact Golgi apparatus. Single cisterns were difficult to differentiate. In contrast, category 2 included bigger “loosened” Golgi complexes. Cisterns were easier to differentiate. Unlike in the previous ones, category 3 Golgi complexes were not closed compartments anymore. They still formed a cisternal structure, but it was possible to differentiate small cisterns/vesicles not being in contact with the main part of the Golgi. The Golgi apparatus in category 4 was fragmented and only this category accounted for the fragmented ones in our statistics in (**d**,**e**). It lost its cohesion and cisternal structure thus became vesicular and the small compartments were dispersed across the cell. (**b**) The structural Golgi changes in control-human iPSC-induced MNs after ALS-CSF application comprised the loss of cisternal configuration and overall Golgi integrity. (**c**) The diagrams depict the distribution of the respective Golgi categories in control- (upper diagram) and mutant-SOD1-MNs (lower diagram) ± CSF treatment. (**d**) The exposition to ALS-CSF caused a significantly higher rate of Golgi fragmentation in WT-human iPSC-induced MNs compared to MNs treated with control-CSF or without CSF. Similar results could be observed comparing SMI32−/TuJ-1+-non MNs treated with ALS-CSF to control-CSF or no CSF-. Of note, SMI32+-MNs exposed to ALS-CSF had a significantly higher Golgi fragmentation rate compared to SMI32−/TuJ-1+-non MNs. (**e**) The application of ALS-CSF to mutant-SOD1 cells caused no significant rate of Golgi fragmentation neither in SMI32+-MNs nor SMI32-/TuJ-1+-non MNs. Data are depicted as mean ± SEM. */**/** *p* < 0.05/0.01/0.001 by one-way ANOVA with Tukey post-hoc test. Scale bar = 5 µm.

**Figure 6 ijms-21-03564-f006:**
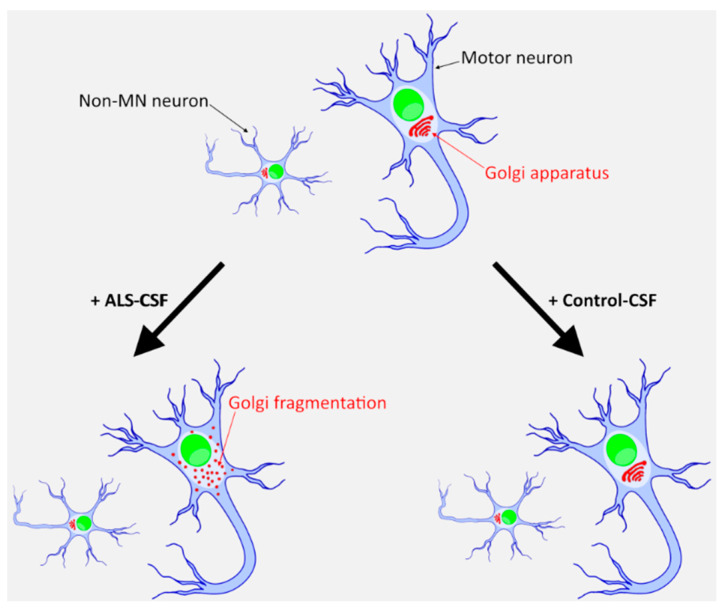
An iPSC-derived neuronal system to model sporadic ALS. Only ALS-CSF, and not healthy donor CSF, induces a degenerative phenotype in healthy donor derived iPSCs. This effect is mainly seen in MNs and only to a much lesser extent in other neuronal subtypes.

**Table 1 ijms-21-03564-t001:** Laboratory values of the used CSF and patient characteristics; data are depicted as mean (SD).

CSF/Patient-Parameter	Control-CSF	ALS-CSF	*p*-Value
Number	8	11	
Gender m:f	5:3	7:4	0.96 ^1^
Age at lumbal puncture, years	47 (21.4)	63 (12.2)	0.08 ^2^
ALS disease onset, spinal:bulbar	n.a.	5:6	
ALS-FSR-R at date of lumbal puncture	n.a.	40.36 (3.67)	
ALS genetic, sporadic:familiar	n.a.	11:0	
Total cell count, MPt/L	1.63 (0.74)	1.27 (0.65)	0.21 ^2^
Total protein, mg/L	412.88 (311.13)	469.55 (223.75)	0.23 ^2^
Albumin, mg/L	308.13 (249.13)	312.82 (148.51)	0.41 ^2^
Glucose, mmol/L	3.59 (0.41)	4.11 (0.92)	0.11 ^2^
Lactate, mmol/L	1.52 (0.22)	1.77 (0.31)	0.10 ^2^
Intrathecal IgG production, yes:no	0:8	0:11	n.a.
Oligoclonal bands, yes:no	1:7	0:11	n.a.
Blood-CSF-barrier dysfunction, yes:no	1:7	3:8	0.44 ^1^

n.a.: not available; ^1^: analysis was performed using *χ*^2^ test; ^2^: analysis was performed using Mann Whitney test.

**Table 2 ijms-21-03564-t002:** Patient/proband characteristics.

Genotyp	Cell Culture Model	Sex	Age at Biopsy (Years)	Mutation	Family History	Age at Disease Onset	Clinical Phenotype	Disease Duration (Months)
controls	hiPSC							
		female	48	-		-	-	-
		male	60	-		-	-	-
		female	45	-		-	-	-
		female	50	-		-	-	-
FUS-ALS	hiPSC							
		female	58	p.R521C	Pos. for ALS	57	spinal	7
SOD1-ALS	hiPSC	male	59	p.R115G	Pos. for ALS	n.a.	spinal	n.a.
TDP-43-ALS	hiPSC	female	87	p.S393L	Pos. for ALS	n.a.	bulbar	n.a.

n.a.: not available.

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
