# Peer review of "Human Spinal Motor Neurons Are Particularly Vulnerable to Cerebrospinal Fluid of Amyotrophic Lateral Sclerosis Patients"

_ijms, 2020, doi:10.3390/ijms21103564_

Round 1

Reviewer 1 Report

Amyotrophic lateral sclerosis (ALS) is a devastating neurodegenerative condition caused by the combinatorial impact of genetics and the environment, with mutations in several genetic loci clearly linked to the disease (e.g. FUS, TDP-43, C9ORF72). Upper and lower motor neurons are the principal cell types affected by ALS, with additional brain pathologies observed in a large proportion of patients, likely due to the fact that ALS lies on a spectrum with frontotemporal dementia. ALS is hypothesised to be prion-like in its progression, with muscle weakness often being described as first presenting in a focal manner with subsequent spreading from this point; cellular models corroborate this idea of cell-to-cell spread. The vast majority of ALS cases are sporadic in nature, which severely impacts our ability to model aspects of this disease, and new techniques are required to improve this situation.

Bräuer and colleagues present a study that assesses the impact of ALS patient cerebrospinal fluid (CSF) on wild-type motor neurons differentiated from human induced pluripotent stem cells (iPSCs). By applying CSF to human motor neurons in culture, the authors have attempted to identify a relevant model for sporadic ALS that relies upon the ALS-prion hypothesis. The only ALS patient CSF-specific phenotype reported is an increase in Golgi fragmentation in motor neurons and other neuronal cells.

Unfortunately, the manuscript has several major flaws that require attention before the work can be considered for publication. Moreover, given the limited useful ALS-specific findings presented in the manuscript, my enthusiasm for the work is equally low. My issues are as follows:

Major

  1. While motor neurons express SMI-32, it is not a selective marker, and really should have been combined or replaced with something more specific. If you have evidence to the contrary, please provide. Alternatively, discuss this as a possible weakness of the work.
  2. The data presented in Figure 2 are suggested to indicate that CSF (both control and ALS patient) reduces viability/number of Tuj1+ neurons relative to Hoechst+ cells (panel C), i.e. CSF has a toxic effect specific to neurons. A similar pattern (although not significant) is also observed for SMI32+ neurons (panel B). Given that CSF is reported earlier in the manuscript to increase proliferation of neural progenitor cells, the authors need to provide more convincing data that the apparent decrease in neuronal cell number is indeed due to loss of neurons and not simply an increase in non-neuronal, Hoechst+ cells.
  3. While the mutSOD1 neurons treated with no CSF may be significantly different from the CSF-treated neurons in Figure 2B (right hand side of the graph), the differences between the mutSOD1 and control cells (found on the left) are non-existent (both control and patient CSF-treated). The authors should therefore caveat their comments on the specific impact of CSF on the ALS-patient motor neurons (lines 118-122), i.e. while the statistics suggest significance, it does not appear to be biologically relevant.
  4. The data presented in Figure 3 are minimal and barely discussed in the text. The cells are reported as having “no overt difference in axon length…” (line 139). This statement is flawed because, A) one cannot tell length of axons from these images, B) no attempt at quantification is attempted, C) no axonal marker is used, and D) no indication is provided in the legend of how many times this “result” has been repeated. Please also include information at what stage these cells have been imaged. The Discussion refers to “neurite degeneration” (line 216), but again, this was not assessed.
  5. No evidence for TDP-43, FUS, or SOD1 inclusions could be identified in the neurons (Figure 4). However, it is unclear whether such inclusions have been observed in iPSC-derived motor neurons. Please provide evidence that this cell type can indeed display inclusions of all three of the highlighted proteins. Also, please include in the legend how many times this experiment was repeated.
  6. The authors helpfully categorise the Golgi into four different types (Figure 5A), yet present data of just category 4 in Figure 5C. Please also include the breakdown of each Golgi category, so the reader can get a better understanding of how the phenotype may be progressing and/or differ between treatments.
  7. The fact that the mutSOD1 motor neurons in Figure 5C do not show a Golgi fragmentation phenotype is concerning for its importance/relevance, please discuss. Also, these data are not at all mentioned in the text.
  8. The Discussion ends suggesting that the Golgi fragmentation is specifically harmful for human iPSC-derived motor neurons (line 258); however, no evidence is provided that the fragmentation is indeed harmful to the cells – their axons show no blebbing or apparent reduction in length (but see point 4) and there is no reduction in cell numbers. For the same reason, I take issue with the statement in the abstract identifying, “golgi fragmentation as early event in the neurodegenerative cascade.”
  9. The title states that ALS CSF selectively affects spinal motor neurons. The only data hinting at this strong conclusion is presented in Figure 5C. However, even these data indicate that SMI-32 negative, Tuj1+ neurons also show a significant increase in Golgi fragmentation compared to control CSF. The title is therefore misinformative.

Minor

  1. The authors continually misuse and chop and change their asterisks for significance, e.g. for Figure 2B, *** is P = 0.0129 and **** P = 0.0411. Please standardise to the norm of * P < 0.05, ** P < 0.01, *** P < 0.001.
  2. Additional information (to age and sex) is needed on the ALS patients from which the serum was extracted (Sporadic? Genes?)
  3. Please confirm the treatment schedule/timepoints used for the images in Figure 1C.
  4. Relating to Figure 2: panel C is discussed in the Results text prior to panel B, which is then referred to before panel A. I would therefore suggest ton rearrange the panels to reflect the order in which they are discussed.
  5. Also, it would greatly help the reader for Figure 2, and all other figures, if the Results text could be linked to each individual figure panel. Make it easy for the reader, rather than them having to try to find all the relevant data.
  6. Figure 6 suggests that ALS-CSF induces degenerative phenotypes, but I can only discern one, both from the data and the graphic.
  7. A thorough proof-read is required.

Author Response

Reviewer 1

Amyotrophic lateral sclerosis (ALS) is a devastating neurodegenerative condition caused by the combinatorial impact of genetics and the environment, with mutations in several genetic loci clearly linked to the disease (e.g. FUS, TDP-43, C9ORF72). Upper and lower motor neurons are the principal cell types affected by ALS, with additional brain pathologies observed in a large proportion of patients, likely due to the fact that ALS lies on a spectrum with frontotemporal dementia. ALS is hypothesised to be prion-like in its progression, with muscle weakness often being described as first presenting in a focal manner with subsequent spreading from this point; cellular models corroborate this idea of cell-to-cell spread. The vast majority of ALS cases are sporadic in nature, which severely impacts our ability to model aspects of this disease, and new techniques are required to improve this situation.

Bräuer and colleagues present a study that assesses the impact of ALS patient cerebrospinal fluid (CSF) on wild-type motor neurons differentiated from human induced pluripotent stem cells (iPSCs). By applying CSF to human motor neurons in culture, the authors have attempted to identify a relevant model for sporadic ALS that relies upon the ALS-prion hypothesis. The only ALS patient CSF-specific phenotype reported is an increase in Golgi fragmentation in motor neurons and other neuronal cells.

Unfortunately, the manuscript has several major flaws that require attention before the work can be considered for publication. Moreover, given the limited useful ALS-specific findings presented in the manuscript, my enthusiasm for the work is equally low. My issues are as follows:

Major

  1. While motor neurons express SMI-32, it is not a selective marker, and really should have been combined or replaced with something more specific. If you have evidence to the contrary, please provide. Alternatively, discuss this as a possible weakness of the work.

Response: We agree with the reviewer, that per se SMI32 is not specific for motor neurons. However, using this specific differentiation protocol, SMI32 seems to be specific for motor neurons in that given neuronal population. This is published in several reports (e.g. PMID: 23533608; or our own previous studies: PMID: 29362359; PMID: 29630989). Nevertheless, we discussed this in the reviesed version of the manuscript.

  1. The data presented in Figure 2 are suggested to indicate that CSF (both control and ALS patient) reduces viability/number of Tuj1+ neurons relative to Hoechst+ cells (panel C), i.e. CSF has a toxic effect specific to neurons. A similar pattern (although not significant) is also observed for SMI32+ neurons (panel B). Given that CSF is reported earlier in the manuscript to increase proliferation of neural progenitor cells, the authors need to provide more convincing data that the apparent decrease in neuronal cell number is indeed due to loss of neurons and not simply an increase in non-neuronal, Hoechst+ cells.

Response: We agree with the reviewer that in the previous version we might not have been clear enough. It is true that we can’t distinguish whether the differences in Fig 2a &b (former 2b & 2c) are caused by proliferating NPCs or neuronal cell death. Since 2c (former 2a) is not affected, it points towards a proliferation effect rather than neuronal cell death. We state this now clearer in the revised manuscript.

  1. While the mutSOD1 neurons treated with no CSF may be significantly different from the CSF-treated neurons in Figure 2B (right hand side of the graph), the differences between the mutSOD1 and control cells (found on the left) are non-existent (both control and patient CSF-treated). The authors should therefore caveat their comments on the specific impact of CSF on the ALS-patient motor neurons (lines 118-122), i.e. while the statistics suggest significance, it does not appear to be biologically relevant.

Response: We agree with the reviewer that even though statistically significant, the differences between WT and SOD1mt neurons are most likely not biological relevant. We corrected our statments in this in the revised version of the manuscript.

  1. The data presented in Figure 3 are minimal and barely discussed in the text. The cells are reported as having “no overt difference in axon length…” (line 139). This statement is flawed because, A) one cannot tell length of axons from these images, B) no attempt at quantification is attempted, C) no axonal marker is used, and D) no indication is provided in the legend of how many times this “result” has been repeated. Please also include information at what stage these cells have been imaged. The Discussion refers to “neurite degeneration” (line 216), but again, this was not assessed.

Response: We agree with the reviewer and now provide now wuantification of neurite network and degeneration (see figure 3(b)) and results part and materials & methods section. As stated in Figure 1(a) we performed all analysis of specific ALS-CSF effects in each experiment at the same timepoint, namly at day 16 (10+6) after reseeding for motor neuron differentiation. We tried to point this out more clearly in the revised version of the manuscript. Additionaly

  1. No evidence for TDP-43, FUS, or SOD1 inclusions could be identified in the neurons (Figure 4). However, it is unclear whether such inclusions have been observed in iPSC-derived motor neurons. Please provide evidence that this cell type can indeed display inclusions of all three of the highlighted proteins. Also, please include in the legend how many times this experiment was repeated.

Response: We thank the reviewer for this concern which is maybe on the heart of disease modelling using iPSCs. Indeed, we and others previously published that cytoplasmic mislocalization and aggregation formation is commonly seen in FUS-ALS models (PMID: 29358088; PMID: 30937520;  or our own previous studies: PMID: 29362359) and has also been reported for TDP-43 (PMID: 30442180, PMID: 31272829). On the contrary, this was never described for SOD1 or C9ORF72 models in a similar way and is clearly known in the field of iPSC workers (e.g. PMID: 29630989). Thus even though using the CSF treatment we could not induce cytoplasmic aggregation of SOD1 and/or TDP43. Noteworthy, it however did not show up as well in FUS-ALS (the more or less positive control). We clarified this in the revised version of the manuscript.

  1. The authors helpfully categorise the Golgi into four different types (Figure 5A), yet present data of just category 4 in Figure 5C. Please also include the breakdown of each Golgi category, so the reader can get a better understanding of how the phenotype may be progressing and/or differ between treatments.

Response: We appreciate this input and added diagrams showing the distribution of the Golgi categorys in the revised figure 5.

  1. The fact that the mutSOD1 motor neurons in Figure 5C do not show a Golgi fragmentation phenotype is concerning for its importance/relevance, please discuss. Also, these data are not at all mentioned in the text.

Response:  We agree with the reviewer that our statement was maybe misleading. We now try to clarify this in the revised version of the manuscript.

  1. The Discussion ends suggesting that the Golgi fragmentation is specifically harmful for human iPSC-derived motor neurons (line 258); however, no evidence is provided that the fragmentation is indeed harmful to the cells – their axons show no blebbing or apparent reduction in length (but see point 4) and there is no reduction in cell numbers. For the same reason, I take issue with the statement in the abstract identifying, “golgi fragmentation as early event in the neurodegenerative cascade.”

Response:  We appreciate the very constructive review and acknowledge that we might not yet have been clear enough. We already tried to point out throughout the manuscript that Golgi fragmentation is – according to the already cited literature – an early sign of neurodegeneration (see PMID: 8203467; PMID: 16545397; PMID: 9542589; PMID: 24708899; PMID: 16382790; PMID: 8643599) and that this was particularly found previously in ALS model systems (see PMID: 21168498; PMID: 17453633; PMID: 23765103; PMID: 16382790; PMID: 8643599). Together with the fact, that in our previous studies with the identical cell lines and this particular differentiation protocol used in these studies, we were able to clearly define the degenerative cascade in the so cultivated motor neurons (see  e.g. PMID: 29362359; PMID: 29630989; PMID: 30422121; PMID: 32151030). Thus we believe that we can make these statements, but carefully add a note about this in the reviesed version of the manuscript.

  1. The title states that ALS CSF selectively affects spinal motor neurons. The only data hinting at this strong conclusion is presented in Figure 5C. However, even these data indicate that SMI-32 negative, Tuj1+ neurons also show a significant increase in Golgi fragmentation compared to control CSF. The title is therefore misinformative.

Response: We understand this concern and modified the title in the revised mansucript.

Minor

  1. The authors continually misuse and chop and change their asterisks for significance, e.g. for Figure 2B, *** is P = 0.0129 and **** P = 0.0411. Please standardise to the norm of * P < 0.05, ** P < 0.01, *** P < 0.001.

Response: We wanted to be more precise but are happy to provide it in a more easy way.

  1. Additional information (to age and sex) is needed on the ALS patients from which the serum was extracted (Sporadic? Genes?)

Response: We acknowledge this comment and combined this with additional data from the CSF (see reviewer 4) in a new table1.

  1. Please confirm the treatment schedule/timepoints used for the images in Figure 1C.

Response: Images were taken after 6 days of CSF treatment, which started 10 days after the reseeding.

  1. Relating to Figure 2: panel C is discussed in the Results text prior to panel B, which is then referred to before panel A. I would therefore suggest ton rearrange the panels to reflect the order in which they are discussed.

Response: We agree that the order should reflect the mentioning in the main text and rearranged it accordingly.

  1. Also, it would greatly help the reader for Figure 2, and all other figures, if the Results text could be linked to each individual figure panel. Make it easy for the reader, rather than them having to try to find all the relevant data.

Response: We only agree in part with the reviewer, since in most of the journals limited amounts of Figures are allowed and thus the readership is used to have the images in a compact way. Furthermore, it helps getting a first brief overview on the manuscript if one can look at five more or less compact figures with proper figure legend. However, if the editor desires to have these changes done, we are happy to provide these.

  1. Figure 6 suggests that ALS-CSF induces degenerative phenotypes, but I can only discern one, both from the data and the graphic.

Response: We agree with the reviewer and precised our statement.

  1. A thorough proof-read is required.

Response: We performed extensive proof reading.

Reviewer 2 Report

The current paper entitled “Cerebrospinal Fluid of Amyotrophic Lateral Sclerosis Patients Selectively Affects Human Spinal Motor Neurons” describes the effects of healthy and ALS CSF on spinal motor neuron. They measured the motor neuron death, neurite lengths, ALS-associated protein aggregations and the golgi fragmentation. Of these measures, in healthy motor neurons treated with ALS-CSF, the authors observed an effect only on golgi fragmentation. However, when the authors add CSF to mutated SOD1 MN, they did not observe any effects. The authors should discuss this point.

Few moderate concerns are listed below.

The authors clearly defined the window where they could analyze the effect of CSF in vitro on iPSC-MN.

Results & Figures:

Figure 2: What does the percentage of MN after CSF treatment given in the text reflect (44.95 and 45.33%): is it the percentage of nuclei positive for SMI32, or SMI32/Tuj or Tuj versus total Hoechst… In that case the percentage given on MN treated with CSF versus MN untreated culture would be artificially lower as CSF treatment increased the number of PNCs. This does not challenge the conclusion that ALS-CSF and healthy-CSF have the same effect on MN survival. However, can the authors reply/re-assure regarding the analysis of MN death, as it is important to assess the same area per well/petri dish and an area that is also representative of the well/dish (it should be more than one or two pictures, and always across all analysis). Please add these details in Material and Methods.

Figure 2 a and b: Why the number of SMI-32+ is lower than SMI-32+/Tuj1+?

Figure 2a and c: Are all the cell Tuj1+ also SMI-32+ in mutated SOD1 culture?

Figure 4: The high magnification images are somewhat blurred - images could be improved.

Hoechst pictures in (f) and (i) are missing. This would help to determine whether SOD1 staining is cytosolic, as it should be.

Figure 5: Did the authors only use category 4 to do their analysis regarding golgi fragmentation? If yes did they observe changes in categories as well with the treatment? How do the authors interpret the absence of effect of ALS-CSF and healthy CSF on mutated SOD1MN? Do they observe the same effects on other ALS cell lines? Are there some protective factors in the CSF that would counteract the intrinsic effect of mutated SOD1, with a beneficial effect greater than the toxic effect of the CSF?

Methods

  • How many PNCs were seeded and differentiated into MN? Was the ratio CSF/differentiated PNC constant across all the experiments?
  • For the untreated MN, was there any PBS-BSA or something else added in the medium of untreated MN to dilute the growth factor, so that growth factor would be at a similar level in CSF-treated and untreated MN cultures, in order to avoid artefactual effects?
  • For the MN death, how many pictures per well/petri dish were taken? Was it consistent across the analysis to make sure a representative area of each well/petri dish was analyzed?
  • For the image analysis, how many cells were analyzed per replicate?
  • Paragraph 4.2: CSF was centrifugated at what speed for how long, and at what temperature?

Author Response

Reviewer 2

The current paper entitled “Cerebrospinal Fluid of Amyotrophic Lateral Sclerosis Patients Selectively Affects Human Spinal Motor Neurons” describes the effects of healthy and ALS CSF on spinal motor neuron. They measured the motor neuron death, neurite lengths, ALS-associated protein aggregations and the golgi fragmentation. Of these measures, in healthy motor neurons treated with ALS-CSF, the authors observed an effect only on golgi fragmentation. However, when the authors add CSF to mutated SOD1 MN, they did not observe any effects. The authors should discuss this point.

Response: We thank the reviewer for this concern which was similarily brought up by Reviewer 1 (item 7). We agree with the reviewer that our statement was maybe misleading. We now try to clarify this in the revised version by discussing whether this is a biological relevant finding or „only“ statistically insignificiant, since the standard deviation is higher in the mt SOD1.

Few moderate concerns are listed below.

Results & Figures:

  1. Figure 2: What does the percentage of MN after CSF treatment given in the text reflect (44.95 and 45.33%): is it the percentage of nuclei positive for SMI32, or SMI32/Tuj or Tuj versus total Hoechst… In that case the percentage given on MN treated with CSF versus MN untreated culture would be artificially lower as CSF treatment increased the number of PNCs. This does not challenge the conclusion that ALS-CSF and healthy-CSF have the same effect on MN survival. However, can the authors reply/re-assure regarding the analysis of MN death, as it is important to assess the same area per well/petri dish and an area that is also representative of the well/dish (it should be more than one or two pictures, and always across all analysis). Please add these details in Material and Methods.

Response: We appreciate this comment. For each individual experiments (of which we always performed ≥ 3) we always randomly took ~15 images per well/condition from the whole well which resulted in 350 up to 3000 cells of each condition of one experment . We now mention this properly in the revised version of the material and methods section.

  1. Figure 2 a and b: Why the number of SMI-32+ is lower than SMI-32+/Tuj1+?

Response: Because we do show relative amounts, not absolute cell counts.

  1. Figure 2a and c: Are all the cell Tuj1+ also SMI-32+ in mutated SOD1 culture?

Response: This connects with the previous comment, we depict relative amounts (ratio). All SMI32 positive cells were allways also costained for Tuj1, but not vice versa.

  1. Figure 4: The high magnification images are somewhat blurred - images could be improved.

Response: We agree with the reviewer but can’t avoid this due to the following: Our intention was to provide the reader convincing images of the cytoplasm, that there is neither relevant cytoplasmic protein mislocalization nor aggregation. Since the respective proteins are highly abundant in the cell nucleus, this strategy will allways lead to a too high intensity in the nucleus causing these blurring. We hope that this fact is appreaciated by the reviewer.

  1. Hoechst pictures in (f) and (i) are missing. This would help to determine whether SOD1 staining is cytosolic, as it should be.

Response: We agree with the reviewer and are happy to provide those images in the according figure now.

  1. Figure 5: Did the authors only use category 4 to do their analysis regarding golgi fragmentation? If yes did they observe changes in categories as well with the treatment?

Response: Since we defined category 4 as fragmented Golgis, only those were used in the statistics concerning Golgi fragmentation. But we now provide additional data of the other categorys in figure 5 of the revised manuscript as well.

  1. How do the authors interpret the absence of effect of ALS-CSF and healthy CSF on mutated SOD1MN?

Response:  We agree with the reviewer that our statement was maybe misleading. We now try to clarify this in the revised version. It look like that SOD1 cells already have higher golgi fragmentation per se and that further CSF treatment does not change this much more. But it also can be due to the larger variance in the SOD1 lines used in the study, thus we do not want to overemphasize this topic.

  1. Do they observe the same effects on other ALS cell lines?

Response: In the mutant FUS cell line we obseved results similar to the WT-cell lines (Golgi fragmentation-rate in MN: no CSF: 1,65%(0,5); control-CSF: 4%(0,7); ALS-CSF: 15,85% (6,3); Golgi fragmentation-rate in SMI-32-/TuJ-1+ neurons: no CSF: 0,7%(1,06); control-CSF: 1,95%(1,06); ALS-CSF: 4,8% (2,55); data in mean (SD)); but because of NPC proliferation only reached n=2 prior the used CSF samples were empty. Thus we decided not to use novel CSF samples only for compelting these studies and did not report this. and therefore not included these data in the paper.

  1. Are there some protective factors in the CSF that would counteract the intrinsic effect of mutated SOD1, with a beneficial effect greater than the toxic effect of the CSF?

Response: We thank the reviewer for this interesting question. We believe that it is most liekly due to the larger variance in the SOD1 lines used in the study, thus we do not want to overemphasize this topic. (See also comment 7)

Methods

  1. How many PNCs were seeded and differentiated into MN?

Response: We seeded 50.000cells/well (26,316 cells/1cm²).We now state this in the method section.

  1. Was the ratio CSF/differentiated PNC constant across all the experiments?

Response: Yes, we thouroughly considered this in every experiment and stated this in the methods section of the revised manuscript.

  1. For the untreated MN, was there any PBS-BSA or something else added in the medium of untreated MN to dilute the growth factor, so that growth factor would be at a similar level in CSF-treated and untreated MN cultures, in order to avoid artefactual effects?

Response: We did this for the very initial experiments. We diluted the medium of untreated cells with PBS and further varied the concentration of the added factors (GDNF, BDNF, cAMP) in the CSF blended medium from 80% to 100%. However, we found no significant influence of this. Furthermore, CSF contains much less proteins than serum (see also tabel of CSF details).

  1. For the MN death, how many pictures per well/petri dish were taken?

Response: We took at least 15 per well (see above comment 1 to the results and methods part). 

  1. Was it consistent across the analysis to make sure a representative area of each well/petri dish was analyzed?

Response: In all conscience!

  1. For the image analysis, how many cells were analyzed per replicate?

Response: For the generation of the relations we counted 350 to 3000 cells per well/condition in each experiment. For the analysis of Golgi morphology we analysed 140 to 220 neurons per well/condition. Concerning the aggregation analysis we watched at several hundred cells per well/condition. We state this now in the revised version of the manuscript.

  1. Paragraph 4.2: CSF was centrifugated at what speed for how long, and at what temperature?

Response: The CSF was centrifugated at 2100 rpm for 10 minutes at room temperature. We now mention this in the materials and method section of the revised mauscript.

Reviewer 3 Report

The manuscript provides a quite innovative insight is ALS pathogenesis. The authors investigated the effect of healthy control and ALS-CSF on iPSCs-derived MNs. They found golgi apparatus fragmentation by addition of ALS-CSF, but not control-CSF. Intriguingly, they did not find pathological protein aggregation, probably because golgi fragmentation is an early event during the pathogenesis.

Although the manuscript is well written, I have both minor and major concerns.

Minors:

Line 42 Pag. 1: CNS is not defined.

Line 51 Pag. 2: Intrathekal has to be corrected in Intratechal.

Line 60 Pag. 2: Alzheimer and Parkinons disease is incorrect. The correct spelling is Alzheimer’s and Parkinson’s disease.

Line 64 Pag. 2: the reference “van Dis et al., 2014” has to be changed following MDPI guidelines.

Line 118 Pag. 4: The SOD1 mutant cell line was not described in the paragraph 2.1. I think that is very important to specify age and gender.

Images captions are not well written. I suggest to write the letter and then to describe the specific image. In the manuscript the letters are written sometimes before the description, sometimes after the description.

Majors:

Line 80 Pag 2.: The authors wrote “We used human iPSC-derived neural progenitor cells (NPC) or neurons from healthy subjects” but in the manuscript it seems that they differentiated NPCs and not neurons. The authors have to be more precise.

Line 84-85 Pag. 2: The mean ages of the two groups are quite different and have a very high SD. I know that CSF is difficult to obtain, but the authors have to claim that such difference can be a potential issue.

Line 87-92 Pag. 2: The authors wrote “One observation was that CSF, irrespective whether from ALS patients or healthy controls, caused a significant proliferation of remaining NPCs in our neuronal cell culture in a clear dose dependent manner (both depending on the amount of CSF and length of exposition).”. The authors claimed that the proliferation is significant, but they did not show any graph or p-value. Can the authors provide a quantitative analysis of what they claim? For example, they can count cells.

Paragraph 2.4 and Discussion: The authors did not write why golgi fragmentation seems to decrease in mutant SOD1 cell line.  I think that these data do not fit well with the work. Although the authors cannot change the data, it is inacceptable that they did not mention it in the results section (except in the caption) and did not discuss it. The authors have to mention ALL the results and have to discuss them, in particular if they are completely unexpected and do not fit with the work. The authors have to give, at least, an explanation of what they observed.

Author Response

Reviewer 3

The manuscript provides a quite innovative insight is ALS pathogenesis. The authors investigated the effect of healthy control and ALS-CSF on iPSCs-derived MNs. They found golgi apparatus fragmentation by addition of ALS-CSF, but not control-CSF. Intriguingly, they did not find pathological protein aggregation, probably because golgi fragmentation is an early event during the pathogenesis.

Response: We deeply appreciate this very positive overall review.

 Although the manuscript is well written, I have both minor and major concerns.

 Minors:

  1. Line 42 Pag. 1: CNS is not defined.

Response: We corrected this minor error.

  1. Line 51 Pag. 2: Intrathekal has to be corrected in Intrathecal.

Response: We corrected this minor error.

  1. Line 60 Pag. 2: Alzheimer and Parkinons disease is incorrect. The correct spelling is Alzheimer’s and Parkinson’s disease.

Response: Corrected

  1. Line 64 Pag. 2: the reference “van Dis et al., 2014” has to be changed following MDPI guidelines.
  2. Response: Corrected

  1. Line 118 Pag. 4: The SOD1 mutant cell line was not described in the paragraph 2.1. I think that is very important to specify age and gender.

Response: We agree with the reviewer and added the missing information in the revised manuscript.

  1. Images captions are not well written. I suggest to write the letter and then to describe the specific image. In the manuscript the letters are written sometimes before the description, sometimes after the description.

Response: We appreciate this suggestion and rewrote the captions accordingly.

Majors:

  1. Line 80 Pag 2.: The authors wrote “We used human iPSC-derived neural progenitor cells (NPC) or neurons from healthy subjects” but in the manuscript it seems that they differentiated NPCs and not neurons. The authors have to be more precise.

Response: This might be a misunderstanding. If not stated different, we always analysed neurons differentiated from iPSC-derived NPCs as it is depicted in Figure 1a. We precised our statment in the revised manuscript.

  1. Line 84-85 Pag. 2: The mean ages of the two groups are quite different and have a very high SD. I know that CSF is difficult to obtain, but the authors have to claim that such difference can be a potential issue.

Response: We agree with the reviewer that there is a larger range however the groups do not differ significantly.

  1. Line 87-92 Pag. 2: The authors wrote “One observation was that CSF, irrespective whether from ALS patients or healthy controls, caused a significant proliferation of remaining NPCs in our neuronal cell culture in a clear dose dependent manner (both depending on the amount of CSF and length of exposition).”. The authors claimed that the proliferation is significant, but they did not show any graph or p-value. Can the authors provide a quantitative analysis of what they claim? For example, they can count cells.

Response: We appreciate this comment and added the missing data as a diagram in Figure 1. We show now that CSF indeed caused a significant proliferation of NPCs depending on CSF-concentration and duration of exposure.

  1. Paragraph 2.4 and Discussion: The authors did not write why golgi fragmentation seems to decrease in mutant SOD1 cell line.  I think that these data do not fit well with the work. Although the authors cannot change the data, it is inacceptable that they did not mention it in the results section (except in the caption) and did not discuss it. The authors have to mention ALL the results and have to discuss them, in particular if they are completely unexpected and do not fit with the work. The authors have to give, at least, an explanation of what they observed.

Response: We agree with the reviewer (and all oher reviewers) on that point and do now thouroughly discuss this in the revised version of the manuscript.

Reviewer 4 Report

The authors deal with an intriguing topic, which is the impact of cerebrospinal fluid (CSF) from amyotrophic lateral sclerosis (ALS) patients on human patient-derived motor neurons (MNs). By using induced pluripotent stem cell-derived MNs from healthy controls and monogenetic forms of ALS, the authors observed a harmful effect of ALS-CSF on healthy donor-derived human MNs, in terms of Golgi fragmentation induced by the addition of ALS-CSF but not control-CSF. They therefore discuss a potentially new way to model early features of sporadic ALS.

Overall, the study is nicely conceived and designed; the results seem to be consistent and are sufficiently discussed. However, there are several concerns needing attention and revision.

MAJOR

- Title: at a first glance, it does not sound clear; what does it mean “CSF affects?” Please rephrase.

- Abstract: it should be extensively revised; e.g. better description of the methods and results; hypothetic mechanisms underlying them; translational value and clinical implications.

- Introduction: in the last paragraph of this section, the authors anticipated some results, which should actually stay in the appropriate section; please move or remove. Additionally, the aim and the experimental hypothesis of the study should be clearly stated.

- Methods: how was the diagnosis of ALS established (revised El Escorial criteria?); please specify. Inclusion/exclusion criteria and ethical requirements are also needed.

- Methods: the reason why the “control subjects” underwent CSF analysis is not clear; I wonder whether they actually were “controls” rather than patients presumably affected by (or suspected for) other neurological diseases. In this context, clinical (i.e. medical history and neurological exam) and instrumental data (i.e. brain-spine MRI and electrophysiology) should be provided (a Table would be helpful).

- Results: a complete report of CSF analysis and comparison between patients and controls are needed for a more insightful evaluation and better understanding of the present study (maybe by using a second Table).

- Results: it is stated that “we had to pool the CSF samples from controls and ALS patients, thus did all further analysis in pooled samples focusing on group differences.” Although I can accept it, on the other hand I wonder whether this might have affected the statistical reliability and reproducibility of the results.

- Discussion: the fact that CSF of ALS patients seems to selectively affect human spinal motor neuron does not fit with the pathophysiology and clinical features of ALS patients, who typically exhibit signs and symptoms of diffuse MNs degeneration (i.e. cortical MNs, spinal MNs, and MNs of the cranial nerves nuclei). I would expect such a selective involvement of spinal MNs in other MN diseases, i.e. spinal muscular atrophy. How and why do the authors explain this?

- Discussion: the integration of results with clinical neurophysiology (i.e. electromyography for the assessment of spinal MN and motor evoked potentials for cortical MN) would have disclosed additional findings. Indeed, clinical-instrumental-CSF correlations are needed, as the authors themselves questioned “whether single ALS patient-derived CSF might vary depending on e.g. clinical parameters”.

- Discussion: please expand the limitation section with all the above-mentioned critical aspects that cannot adequately addressed in the present investigation.

MINOR

- General: please replace “golgi” with “Golgi” throughout the manuscript.

- Introduction: please briefly explain what the “NSC34 cultures” are.

- Introduction: please replace “Intrathekal” with “Intratechal”; few words after: “in vivo” with “in vivo”.

- Introduction: please replace “reviewed in [15]” with “reviewed by Sundaramoorthy and colleagues [15]”.

- References: please include “(van Dis et al., 2014)” and “(Buddensiek et al., 2010)” within the reference list.

Author Response

Reviewer 4

The authors deal with an intriguing topic, which is the impact of cerebrospinal fluid (CSF) from amyotrophic lateral sclerosis (ALS) patients on human patient-derived motor neurons (MNs). By using induced pluripotent stem cell-derived MNs from healthy controls and monogenetic forms of ALS, the authors observed a harmful effect of ALS-CSF on healthy donor-derived human MNs, in terms of Golgi fragmentation induced by the addition of ALS-CSF but not control-CSF. They therefore discuss a potentially new way to model early features of sporadic ALS.

Overall, the study is nicely conceived and designed; the results seem to be consistent and are sufficiently discussed.

Response. We deeply acknowledge this positive review.

However, there are several concerns needing attention and revision.

MAJOR

  1. Title: at a first glance, it does not sound clear; what does it mean “CSF affects?” Please rephrase.

Response: We modified the title appropriately (see also Reviewer 1)

  1. Abstract: it should be extensively revised; e.g. better description of the methods and results; hypothetic mechanisms underlying them; translational value and clinical implications.

Response: We modified the Abstract accondingly, however also needed to stick to the journal‘s guidelines. Finally, we do not want to make wrong attributions towards clinical translation/implication, since the intention of our study was to investigate whether ALS-CSF can be used to generate an in vitro model system for sporadic ALS.

  1. Introduction: in the last paragraph of this section, the authors anticipated some results, which should actually stay in the appropriate section; please move or remove. Additionally, the aim and the experimental hypothesis of the study should be clearly stated.

Response: We modified it as wished.

  1. Methods: how was the diagnosis of ALS established (revised El Escorial criteria?); please specify. Inclusion/exclusion criteria and ethical requirements are also needed.

Response: We added this missing data.

  1. Methods: the reason why the “control subjects” underwent CSF analysis is not clear

I wonder whether they actually were “controls” rather than patients presumably affected by (or suspected for) other neurological diseases

Response: The control group consisted of a different neurological diseases. These included patients with ischemic strokes (one caused by a persisting forman ovale and the other through a cardioembolic genesis [both lumbal punctures were performed to exclude inflammatory genesis]). Further patients were diagnosed with primary headache disorders. Another patient was diagnosed with vascular dementia and the lumbal puncture was performed to exclude Alzheimer disease (tau level and beta amyliod ratio normal). The last patient suffered from a cervical spinal stenosis. The operation took place 7 years earlier and the deficits were residual. The lumbal puncture was performed to exclude a secondary inflammatory process. All in all the control group consisted of subjects with neurological disease, which should not influence the question raised in the current study. Since it is very difficult to obtain CSF from „absolutly“ healthy subjects, this was as close as we could get. In addition, since ALS is an old age disease, almost no patient does not suffer from additional other common disease as well (e.g. headache, vascular leasions of the brain, etc.). We mentonoed this in the Methods section.

  1. In this context, clinical (i.e. medical history and neurological exam) and instrumental data (i.e. brain-spine MRI and electrophysiology) should be provided (a Table would be helpful).

Response: We now show more details on some basic characterization of the patients and controls including CSF details, however do think that mentioning all details for the controls are missleading and not the scope of this manuscript. We exclude every CSF sample with a relevant clinical or paraclinical sign of inflammation and/or degenerative hallmarks and only include „neurological“ patients in which teh CSF was performed excluding relevant neurological disease.

  1. Results: a complete report of CSF analysis and comparison between patients and controls are needed for a more insightful evaluation and better understanding of the present study (maybe by using a second Table).

Response: We added this in a new table 1.

  1. Results: it is stated that “we had to pool the CSF samples from controls and ALS patients, thus did all further analysis in pooled samples focusing on group differences.” Although I can accept it, on the other hand I wonder whether this might have affected the statistical reliability and reproducibility of the results.

Response: We believe that pooling the CSF is even more helpful to avoid artificial variation due to single patients (family, epigenetic, genetic) background. We also did not want to look at influencing factors within ALS patients but whether sporadic ALS patients‘ CSF does induce motor neuron disease mimics in the dish.

  1. Discussion: the fact that CSF of ALS patients seems to selectively affect human spinal motor neuron does not fit with the pathophysiology and clinical features of ALS patients, who typically exhibit signs and symptoms of diffuse MNs degeneration (i.e. cortical MNs, spinal MNs, and MNs of the cranial nerves nuclei). I would expect such a selective involvement of spinal MNs in other MN diseases, i.e. spinal muscular atrophy. How and why do the authors explain this?

Response: This was obviously a missunderstanding. We wanted to state that within the shown model system of a spinal cord neuronal culture, the main affected neurons by ALS CSF are the motor neurons. We did not wanted to state this as a general statement including all mentioned nerve cells, since we did not adress this experimentally in our study. We tried to point this out more clearly.

  1. Discussion: the integration of results with clinical neurophysiology (i.e. electromyography for the assessment of spinal MN and motor evoked potentials for cortical MN) would have disclosed additional findings. Indeed, clinical-instrumental-CSF correlations are needed, as the authors themselves questioned “whether single ALS patient-derived CSF might vary depending on e.g. clinical parameters”.

Response: As mentioned above, the intention was not to investigate „whether single ALS patient-derived CSF might vary depending on e.g. clinical parameters” but whether ALS-CSF is able to induce a motor neuronal degenerative phenotype in the dish thus enabling disease modelling of sporadic ALS per se. We clarified this throughout the revised mansucript.

  1. Discussion: please expand the limitation section with all the above-mentioned critical aspects that cannot adequately addressed in the present investigation.

Response: We expand this section accordinlgy

MINOR

  1. General: please replace “golgi” with “Golgi” throughout the manuscript.

Response: We corrected this aspect.

  1. Introduction: please briefly explain what the “NSC34 cultures” are.

Response:

  1. Introduction: please replace “Intrathekal” with “Intratechal”; few words after: “in vivo” with “in vivo”.

Response: We corrected these minor errors.

  1. Introduction: please replace “reviewed in [15]” with “reviewed by Sundaramoorthy and colleagues [15]”.

Response: We improved this formulation as proposed.

  1. References: please include “(van Dis et al., 2014)” and “(Buddensiek et al., 2010)” within the reference list.

Response: We included these references.

Round 2

Reviewer 1 Report

I congratulate the authors on a successful revision. The manuscript is now of sufficient quality for publication. However, I have one reservation that needs attending to - the new title does not read well. I suggest a few of alternatives that the authors should consider:

Human spinal motor neurons are particularly vulnerable to cerebrospinal fluid of amyotrophic lateral sclerosis patients (this is a slight correction of the authors' title)

or

Human iPSC motor neurons are particularly vulnerable to ALS patient cerebrospinal fluid

or

ALS patient cerebrospinal fluid preferentially induces Golgi fragmentation in human iPSC motor neurons

I prefer the last one, but totally leave the decision up to the authors.

Well done

Author Response

I congratulate the authors on a successful revision. The manuscript is now of sufficient quality for publication.

Response: We deeply appreciate this statement.

However, I have one reservation that needs attending to - the new title does not read well. I suggest a few of alternatives that the authors should consider:

Human spinal motor neurons are particularly vulnerable to cerebrospinal fluid of amyotrophic lateral sclerosis patients (this is a slight correction of the authors' title)

or

Human iPSC motor neurons are particularly vulnerable to ALS patient cerebrospinal fluid

or

ALS patient cerebrospinal fluid preferentially induces Golgi fragmentation in human iPSC motor neurons

Response: We adopted the title and tried to find a compromise also with reviewer 4.

Reviewer 2 Report

The authors answered all my comments. The paper is now improve.

I would have a very small comment to add:

check in Figure 5c: the stat should be P<0.01 instead of P<0,01... idem for the other stats in the figure.

I would also suggest to change "categories" but the actual description of the Golgi structure, to ease the reading of the figure. 

Author Response

The authors answered all my comments. The paper is now improve.

Response: We deeply appreciate this statement.

I would have a very small comment to add:

check in Figure 5c: the stat should be P<0.01 instead of P<0,01... idem for the other stats in the figure.

Response: We corrected this mistake.

I would also suggest to change "categories" but the actual description of the Golgi structure, to ease the reading of the figure. 

Response: We understand the issue of the reviewer, however found no fitting keywords describing best the single categories. We believe that the image in figure 5a are the best explanations about the different cotegories and thus want to leave it as it is..

Reviewer 3 Report

The authors replied to all my concerns and improved the quality of the manuscript. I do not have any other concern about the manuscript.

Author Response

The authors replied to all my concerns and improved the quality of the manuscript. I do not have any other concern about the manuscript.

Response: We deeply appreciate the positive review.

Reviewer 4 Report

The authors have adequately addressed most of my concerns, thus improving the quality of this manuscript. I still have a couple of remaining comments:

- Title: please replace it with the following: “Human spinal motor neurons are particularly vulnerable to cerebrospinal fluid of amyotrophic lateral sclerosis patients”

- Discussion: although I share the authors’ explanations, the fact that the control group consisted of patients with current or past neurological diseases should be listed among the limitations of the study. Similarly, given that the observed effects were not tested on all types of motor neurons but on spinal motor neurons only, and since different types of motor neuron diseases do exist (ALS, primary LS, spinal, SMA, etc.), the lack of generalization of the present findings should be clearly stated.

Author Response

The authors have adequately addressed most of my concerns, thus improving the quality of this manuscript.

Response: We deeply appreciate the positive review.

- Title: please replace it with the following: “Human spinal motor neurons are particularly vulnerable to cerebrospinal fluid of amyotrophic lateral sclerosis patients”

Response: We adopted it accodingly (see also Reviewer 1)

- Discussion: although I share the authors’ explanations, the fact that the control group consisted of patients with current or past neurological diseases should be listed among the limitations of the study.

Response: We appreciate this comment and add this tot he revised discussion.

-Similarly, given that the observed effects were not tested on all types of motor neurons but on spinal motor neurons only, and since different types of motor neuron diseases do exist (ALS, primary LS, spinal, SMA, etc.), the lack of generalization of the present findings should be clearly stated.

Response: We appreciate this comment. We’d like to point out that we already had the following sentences in the discussion:

„Furthermore, the model system presented represents one aspect of ALS (namely a model for spinal cord degeneration), thus there might be differences when analyzing other neuronal cell populations affected, e.g. cortical neurons. „

We do believe (and the other 3 reviewers obviously as well) that this is sufficiently discussed. The differences between different subtypes of motor neuron disease was and is not part of the manuscript and thus stands outside the focus of the manuscript. Additionally, we believe that the mentioned details are basic knowledge in the field.